# Biocontrol Potential of Some Rhizospheric Soil Bacterial Strains against *Fusarium culmorum* and Subsequent Effect on Growth of Two Tunisian Wheat Cultivars

**DOI:** 10.3390/microorganisms11051165

**Published:** 2023-04-29

**Authors:** Habiba Kouki, Mouna Souihi, Ilhem Saadouli, Sabrine Balti, Amira Ayed, Nihed Majdoub, Amor Mosbah, Ismail Amri, Yassine Mabrouk

**Affiliations:** 1Laboratory of Biotechnology and Nuclear Technology, National Centre for Nuclear Sciences and Technologies (CNSTN), Sidi Thabet, Technopark, Ariana 2020, Tunisia; habibakouki96@gmail.com (H.K.);; 2Faculty of Sciences of Bizerte, Carthage University, Jarzouna 7021, Tunisia; 3Laboratory of Microorganisms and Active Biomolecules, Faculty of Sciences of Tunis, University of Tunis El Manar, Tunis 2092, Tunisia; 4Higher Institute of Biotechnology of Sidi Thabet (ISBST), University of Manouba, Ariana 2020, Tunisia; 5Laboratory of Biotechnology and Valorization of Bio-Geo Resources, Higher Institute of Biotechnology of Sidi Thabet (ISBST), University of Manouba, Ariana 2020, Tunisia

**Keywords:** biocontrol, *Fusarium culmorom*, rhizobacteria, PGPR traits, wheat

## Abstract

PGPR (Plant Growth Promoting Rhizobacteria) are used as biofertilizers and biological control agents against fungi. The objective of this work was to evaluate the antagonistic activities of some bacterial strains isolated from soil against four phytopathogenic fungal strains (*Fusarium graminearum*, *F. culmorum*, *Phytophthora* sp. and *Verticillium dahlia*). Two strains having an antagonist effect on fungi and displaying the maximum of plant growth promoting (PGP) traits were selected for further study and identified as *Bacillus subtilis* and *B. amyloliquefaciens* respectively. *In planta* assays demonstrated that the two *Bacillus* strains are able to enhance plant growth of two wheat cultivars in absence of nitrogen and protect them against *F. culmorum*. Pot experiments performed in a greenhouse showed that wheat plants inoculation with two bacterial strains reduce *F. culmorum* disease severity correlated with the accumulation of phenolic compounds and chlorophyll content. These could partly explain the effectiveness of these bacteria in protecting Tunisian durum wheat cultivars against *F. culmorum*. Application *B. amyloliquefaciens*, showed better protection than *B. subtilis* although the last one enhanced more the plant growth of two wheat cultivars in absence of fungus. Hence, combination of two bacterial strains could be a strategic approach to enhance plant growth and control plant diseases.

## 1. Introduction

In Tunisia, wheat cultivation is economically important. It represents approximately 60% of cereal crops. The majority of these crops are limited by insufficient and irregular rains. The low rainfall conditions predispose this crop to root rot, a fungal disease that is widely responded to in cereal-growing regions and causes significant damage. Some researchers showed that *F. culmorum* is the main pathogen involved in this disease in Tunisia [1]. Chemical control has proven ineffective or not very effective against *Fusarium* sp. [2]. As a result, cultural practices and biological control are the best means of combating wheat root rot.

Nevertheless, the excessive use of fungicides for the treatment of plants leads to the emergence of resistance in pathogenic fungal strains [3]. In addition, negative effects can be observed on human and animal health following the use of synthetic fungicides to control these diseases [4]. To overcome these limitations, the search for alternatives for the sustainable management of wheat diseases has become necessary. Antagonistic bacteria protect plants against attacks by phytopathogens hence protect them against various diseases [5]. The use of rhizosphere bacteria as biofertilizers or as agents for the biological control of pathogens is considered an alternative to polluting chemicals in the environment [6]. Bacteria can promote plant health and yield by promoting plant growth through many different processes [7]. Among these processes we cite the fixation of atmospheric nitrogen; solubilization of unavailable nutrients; altered root system development [8]; enhancing of genes involved in plant resistance or acquisition of nutrient; antagonistic activity against pathogenic soil microorganisms [9]; and exudation of metabolites with antibiotic activities [10].

In order to better enhance the beneficial effect of rhizobacteria, we proposed to explore, in this work, the antifungal activities and PGP traits of some bacterial strains obtained from the wheat rhizosphere. Firstly, we have selected two bacterial isolates S6 and S8 based on their antagonistic power against fungal strains and presenting the maximum of PGP traits and then identified these bacterial isolates using biochemical and molecular approaches. Finally, we have evaluated their capacities to enhance plant growth and protect two Tunisian durum wheat cultivars (Bouanene cv and Hydra cv) against *F. culmorum* in pots under controlled conditions by measuring, plantlet growth, disease severity, phenolic and cholophyll contents.

## 2. Materials and Methods

Bacterial strains were isolated as described by Travers et al. [11] using soil taken from the rhizosphere of plants. Then, 1 g of soil was suspended in sterilized dH_2_O (10 mL) after which 1 mL of each suspension was mixed with 10 mL of LB nutrient broth (Luria Bertani) at pH 6.8. Repeated subcultures of individual colonies were carried out in order to purify the isolates of bacteria.

### 2.1. Antagonistic Activity of Bacterial Isolates In Vitro against Fungal Strains

A preliminary screening test was conducted using the agar disc method on PDA medium to select the isolates having an antagonistic effect from collection of bacterial isolates. A fungal disk with a diameter of 6 mm was placed in the middle of a 90 mm Petri dish containing PDA medium and a streak of bacterial isolates was positioned at a distance of 2.5 cm from the disk on the same Petri dish [12]. Antagonism was recorded as (+) or (−) as present or absent respectively. Bacterial isolates antagonistic (+) against fungal strains were selected for second-round screening using the in vitro double culture assay [8]. The inhibition of fungal growth was recorded five days after incubation at 24 °C and the percentage of reduction of the growth of the mycelium was calculated according to the formula
$$I=\frac{R1-R2}{R1}\;\times \;100$$
where R1 is the growth of mycelium in the plate without bacteria and R2 is the growth of mycelium towards the bacterial streaks. All experiments were carried out three times with three repetitions each.

### 2.2. PGP Traits of Bacterial Strains

#### 2.2.1. Phosphate Solubilization Assay

The capacity to solubilize phosphate from unsolvable forms (inorganic and organic) was verified according to the method described by [13]. Bacterial isolates were cultured in National Botanical Research Institute’s phosphate growth medium (NBRIP) and phytase-screening medium (PSM) broth, containing Ca-phosphate (Ca_3_(PO_4_)_2_ and Na-phytate (C_6_H_18_O_24_P_6_•xNa+•yH_2_O) as only sources of phosphate (P), respectively. These cultures were incubated for 7 days at 30 °C. After incubation, the cultures were centrifuged at 10,000 rpm for 10 min and orthophosphate (P) released in the supernatant was quantified as described by Bae et al. [14] and expressed in mg/mL.

#### 2.2.2. Nitrogen Fixation

For nitrogen fixation quantification bacterial isolates were incubated on nitrogen-free medium (NFM) for 7 days, then bacterial growth was qualitatively checked [15].

#### 2.2.3. Indole-3-Acetic-Acid Production

The culture of the bacterial isolates was carried out in a medium to which tryptophan was added for 7 days at 28 °C. Then, the bacterial cultures were centrifuged at 10,000 rpm for 10 min followed by the addition of 2 mL of Salkowski’s reagent to the supernatants obtained. The synthesis of IAA is proven by the appearance of the pink color and its quantification is based on the reading of the optical density by a microplate reader (Thermo Scientific™ 138 Multiskan™ FC-MIB 51119000, Vantaa, Finland) at 535 nm using commercial IAA (Sigma Aldrich, St. Louis, MO, USA) as standard [16].

#### 2.2.4. Ammonia Production

Ammonia production was estimated using the method described by abdelwahed et al. [17]. Briefly, cultures of bacterial isolates were carried out in a peptone water broth at 30 ± 2 °C for 7 days of incubation. Then the cultures were centrifuged and the bacterial supernatants were distributed in different wells at the rate of 50 μL per well and 100 μL of Nessler’s reagent are added then the microplate was incubated for 10 min at room temperature. Then, 55 μL of each reaction mixture were transferred into a second 96-well microplate to which 330 μL of ultrapure water is added to obtain a final volume of 385 μL and the optical density is taken at 450 nm. Ammonia concentration was determined using standard ammonium sulfate solution.

#### 2.2.5. Cellulase Detection

Cellulase estimation was carried out qualitatively according to the method described by [18] by point inoculation of the strain on a basic agar medium containing carboxymethylcellulose as the sole carbon source. After 5 days of incubation at 30 °C, all plates were stained with 1% (*w*/*v*) Congo red solution for 15 min then destained with NaCl (1 M) for 15 min. The production of cellulase has been proven by the formation of a zone of degradation around the bacterial colonies.

#### 2.2.6. Siderophore Assay

Quantification of siderophores was performed by Chrome Azurol S (CAS) assay [19,20]. to prepare the CAS solution 72.9 mg (hexadecyl trime-thylammonium bromide (HDTMA) was dissolved in 40 mL of dH_2_O and 60.5 mg (chromium azurol S) in 50 mL of dH_2_O, then mixed together to which one add 10 mL of the Fe^3+^ solution (1 mM FeCl_3_ in 10 mM HCl). Then the CAS solution was sterilized and mixed with autoclaved King’s medium with a ratio of 1:9, respectively. Quantification of siderophores was performed by mixing equal volumes (100 µL) from bacterial culture supernatants and CAS reagents. This was followed by incubation at room temperature for 20 min and absorbance was measured at 630 nm in a microplates reader [21] and the quantities of siderophores was determined in percentage of siderophore units (psu) according to Payne’s method [22].

### 2.3. Identification of Selected Bacterial Isolates

The identification of selected strains was done morphologically by observation under an immersion optical microscope followed by gram staining. Then molecular identification was done using sequencing of the 16S rDNA gene analysis. Genomic DNA was isolated as described by [23] and the extracted DNA was quantified by nanodrop and analyzed qualitatively on agarose gel (1% *w*/*v*). The universal primers fD1 (AGAGTTTG ATCCTGGCTCAG) and rD1 (AAGGAGGT GATCCAGCC) were used for amplification of the 16S rDNA genes. The purified PCR products were then subjected to automated sequencing “https://www.gexbyweb.com (accessed on 24 July 2016)”) and the sequences obtained are subjected to comparison to those on the NCBI and this using the blast program.

### 2.4. PGPR and Protection of Wheat Plants against F. culmorum by Selected Bacterial Strains (S6 and S8)

#### 2.4.1. Wheat Inoculation by Selected Bacterial Strains and Plant Growth Promotion

Seeds of two wheat cultivars (CV Bouanene and CV Hydra) were surface sterilized in 2% of solution of calcium hypochlorite for 30 min and then washed with sterile distilled water. Sterilized seeds were soaked in bacterial inoculum10^8^ cells mL^−1^ of each bacterial strain and then transplanted in plastic pots containing sterilized soil. Plants were provided with an N-free nutrient solution every 7 days [24]. Plants were cultivated in a light 16 h cycle and 25 °C temperature. Dry weight of shoots and roots and length of shoots and roots were recorded after 30 days of cultivation.

#### 2.4.2. Wheat Plants Inoculations with Selected Bacterial Strains and *F. culmorum* and Disease Assessment

The efficacy of two selected bacterial strains (S6 and S8) to reduce the disease incidence of wheat cultivars (CV Bouane and CV Hydra) was assessed in pots in controlled conditions. Sterilized seeds were soaked with bacterial inoculums (10^8^ cells mL^−1^) for 30 min and transplanted in pots containing soil. After 10 days of transplanting, plants were inoculated with 1 mL of *F. culmorum* inoculum at 10^6^ spores mL^−1^. The treatments of this experiment were as follows: plants treated with *F. culmorum* and inoculated with bacteria, plants treated only with *F. culmorum* and noninoculated wheat. Length dry weight of wheat plants (root and shoot) were recorded 30 days after treatments with fungi. Samples were dried in an oven at 60 °C t for 72 h for measuring the dry weight of seedling. symptoms of disease associated with Fusarium plant infection and the virulence of pathogen was evaluated and the recorded and Disease Index (DI, %) was determined using the formula:(1)$$DI\left(\%\right)=(\frac{Ni}{Nt})$$

With Ni, number of infected plants; Nt, total number of plants.

### 2.5. Extraction and Quantification of Phenol and Chlorophyll

Leaves tissues of three plantlets were collected after 0, 24, 48 and 72 h post inoculation with *F. culmorum*. Phenols were quantified using the Folin−Ciocalteu method according to [25] using gallic acid as standard.

For chlorophyll estimation leaves were crushed in acetone (80%) and the absorption was determined at different wavelengths (λ 665, 663, 649, 646, 470 nm). The concentrations of different chlorophyll forms were calculated using the formula proposed by [26].

### 2.6. Statistical Analysis

All presented data were expressed as mean ± SD. Analysis of variance (ANOVA) was done for each set of disease data using the SPSS software. For independent repeated experiments, the data were tested for significant differences. Values with *p* < 0.05 were considered statistically significant.

## 3. Results

### 3.1. Bacterial Isolates Antagonism against Fusarium culmorum and Their Plant Growth-Promoting Characteristics

Forty isolates were obtained from collected soil from rhizosphire and tested for their antagonistic effects against 4 phytopathogenic fungi. Among these 40 bacterial isolates, 9 isolates showed antagonistic activities against the fungal strains studied and all these bacteria inhibited the growth of the mycelium with different percentages depending on the fungal strain tested 7 days after inoculation (Table 1, Figure 1). The six bacterial strains (S1, S5, S6, S7, S8 and S9) allowed the inhibition of the 4 fungal strains tested. S1 showed the highest inhibition ranging from 71% to 87.5% on day 7 post inoculation and bacterial strains S4, S3, S2 and S9 caused the lowest growth inhibition for *F. graminearum*, Verticillium dahlia, *Phytophtora* sp. and *F. culmorum* respectively. An increase in the transparency of the fungus is observed in the plates inoculated with the bacterial strains S6 and S8, while for the uninoculated control, the appearance of the fungal colony is darker and homogeneous throughout the Petri dish (Figure 1D).

In vitro PGP traits results were summarized in Table 2 for the nine selected bacteria. Bacterial isolates (S3, S5, S6, S7 and S8) solubilized mineral phosphate on agar PVK medium. S3, S6 and S8 produced the main solubilization halos and only strain S8 showed organic phosphate mineralization (Table 2, Figure 2). Results resumed in Table 2 clearly showed only S6 is able to produce IAA. All tested bacteria were able to produce ammonia and the quantity of ammonia produced vary between bacterial isolates and values are ranged between 0.002 to 0.69 mg/mL, with the highest production was obtained for S6 and the lowest for S9 (Table 2). The majority of bacterial isolates tested have the ability to fix nitrogen in vitro except for strains S1, S7 and S9. Four strains produced cellulase (S1, S2, S6 and S9). For the siderophore, only S6 and S8 allowed the production justified by the appearance of halos colored yellow and purple around their respective colonies in the O-CAS test (Table 2, Figure 2). Notably, isolates S6 and S8 showed significant antagonistic activity against *F. culmorum* (Table 1) and maximum PGPR traits with siderophore production (Table 2). These two isolates were selected for characterization and study of their effects on two wheat cultivars growth and their protection against Fusarium wilt in pots under controlled conditions.

### 3.2. Characterization and Identification of Isolates S6 and S8

Phenotypic characteristics and 16S rDNA sequence analysis associate the two strains with the Bacillus group. According to 16S rDNA sequence homology, isolate S6 strain is identified as *B. subtilis* subsp. subtilis str. 168 while the S8 strain is characterized as *Bacillus amyloliquefaciens* (Table 3). These two strains are very different for morphology and texture on solid medium, one is rough (S6) and the other is a little soft and very difficult to sample (S8). Both *B. subtilis* and *B. amyloliquefaciens* have shown antifungal activity and therefore have potential for use in biological control. Moreover, these two strains showed PGP traits in vitro. The next step will be to determine the effects of these *Bacillus* strains on wheat plant growth and plant disease development in response to fungal treatment in pots under controlled condition.

### 3.3. Plant Growth Promotion Effect of Selected Bacterial Strains on Two Wheat Cultivars (CV Bouanene and CV Hydra)

Compared to control plants, plants treated with two bacterial strains showed a very significant increase in dry weight of shoots with a greater effect using *Bacillus subtilis* (strain S6) for Hydra wheat cultivar (Figure 3). In particular, the total dry biomass of the aerial parts of the plants inoculated with the S6 and S8 bacteria was higher by approximately 25 to 29% than that of the control plants in the cultivar Bouanene and by approximately 71 to 72% in the cultivar Hydra respectively. A significant increase was also observed for root biomass (30 to 37%) for CV Bouanene and (13 to 14%) for CV hydra inoculated with S6 and S8 respectively. The inoculation of wheat plants with the two bacterial strains allowed an increase in the growth parameters for the two varieties studied (CV Hydra and CV bouanene) and the biomasses obtained are better than those observed in the plants treated with nitrogen. The two strains showed a greater improvement in aerial biomass for the Hydra cultivar, whereas a greater improvement was observed for the roots in the Bouanene cultivar.

### 3.4. Biocontrol of F. culmorum by Two Selected Bacterial Strains

*F. culmorum* affected the growth of the two wheat cultivars differently, and the development of disease symptoms was more pronounced in the Bouanene cultivar compared to the CV Hydra. Following infection with the fungus, a reduction in the dry biomass of the aerial parts of around 50% with a disease incidence of 100% was observed in the CV bouanene, while the reduction in aerial dry biomass does not exceed 20% in the Hydra cultivar with a disease incidence of 50% (Figure 4, Figure 5 and Figure 6). Analysis of the results showed that two bacterial strains were able to significantly suppress the disease severity (Figure 4, Figure 5 and Figure 6). In the plants treated with the strains (*B. subtilis* S6 and *B. amyloliquefaciens* S8), no symptoms of necrosis were observed on the root. The treatment *B. amyloliquefaciens* was the best treatment shoot growth (length and dry weight) compared to the control none infected by fugus. The pathogen reduced the mean shoot dry weight by 35% (Figure 4). The seeds treatments with two bacterial strains significantly improved the shoot dry weight in comparison to the infected control and were in the same statistical group as the healthy control. It was also determined that the pathogen reduced the root dry weight by about 40%. The strain *B. subtilis* S6 was the best treatment and increased root dry weight up to 1.75 times compared with the infected control for Hydra cultivar. In addition to improved plant growth, disease indices showed a significant reduction in the incidence of Fusarium wilt in 30-day-old wheat plants following the application of bacterial strains (S6 and S8) by compared to seedlings without any bacterial treatment. In addition, the *B. amyloliquefaciens* strain was found to be relatively more effective (approximately (95%) wheat protection than the *B. subtilis* bacterial strain (approximately 85% wheat protection).

Examination of chlorophyll uptake showed that application of bacterial strains generally resulted in stimulation of total chlorophyll synthesis. Wheat seedlings treated with the fungal strain had similar chlorophyll contents compared to untreated seedlings (Figure 7).

In response to fungal strain infection, a decrease in phenol content was observed for two cultivars (Bouanene and Hydra) at 24 h and 48 h after infection with a higher decrease in the Hydra cultivar case. While the application of bacterial strains increased the content of phenolic compounds in the leaves of two cultivars (Figure 8).

## 4. Discussion

To manage pathogens in agriculture using biological controls is consider as an effective alternative method avoiding the use of pesticides harmful for the environment accumulating in plants and perturbing composition of beneficial organisms in the soil [27]. Despite the different microorganisms that are used as a control agent, the metabolites produced by the bacteria obtained from the soil contribute to reduction in growth of the phytopathogene fungi [28]. Bacillus has demonstrated broad spectrum plant protection, antifungal potency and use as a growth factor [29,30,31]. This bacterium is of great importance in the biotechnology industry and in agriculture [32]. In our present study, inhibition is discerned by growth limit or complete absence of fungal mycelium using double culture technique when bacterium and fungus grow on PDA medium. Co-culture of the fungus with the different bacterial isolates showed that the majority of isolates reduced fungal growth by forming a zone of inhibition. The decrease in the growth of the pathogen by bacteria isolated from soil observed in vitro, could be explained by the secretion of secondary metabolites of the bacteria. These biomolecules may include antibiotics or lytic enzymes such as chitinases or glucanases. Several investigators have reported the involvement of metabolites produced by rhizobacteria in the biological control of several phytopathogens [30,31,32,33]. On the other hand, a change in the color of the mycelium during co-culture of *F. culmorum* with two *Bacillus* strains was observed. It is darker in the absence of bacteria.

Farmers’ intensive use of inorganic fertilizers has shown harmful effects on natural ecosystems and human health. The use of soil bacteria as bio-fertilizer is helpful for farmers and environmentally friendly way to enhance plant yield and produce safe food in a sustainable agricultural system [34]. In our investigation, we focused on quantifying the effect of bacterial treatments on the growth of wheat plants using potted experiments under controlled conditions. The effect of bacterial inoculations on yields of crops has been reported in previous work using pots and in field conditions [35,36]. The rise in shoot length may be due to molecules produced by some bacterial strains and nutrients mineralization and make them available to plants [37]. Other workers have shown that the application of PGPR increased soil pH, dry and fresh weight of wheat roots [38]. Moreover, the work of saber et al. [39] reported that the use of PGPR could maintain wheat yield related parameters with a significant reduction of the application of *p* and N fertilizers. Results presented in this study also showed that when PGPB was applied without N supplementation in a nutrient solution, the dry weight of wheat was higher than when using N. Similarly, the increase in root length after PGPR inoculation is explained by the production of phytohormones by these bacteria [40,41].

The objective of our study was to use bacteria isolated from soil with potential as inoculants to reduce the application of chemical as fertilizers and pesticides. Results showed that PGP bacteria plays an important role by producing hormone and making the nutrients available. Over the past decades, research on PGP bacteria has demonstrated that these bacterial strains can enhance plant growth by two ways resulting to the production of plant growth hormones such as IAA [42], that it exhibits antagonistic activity against plant pathogens in the soil by producing siderophore [43] and that it causes the solubilization of mineral phosphates with other nutrients [44]. The production of siderophores by bacteria in the rhizosphere is important for plant iron nutrition, resulting in a significant increase in plant growth [45] and their resistance against fungal attacks [46]. Some workers showed that the siderophores produced by *Pseudomonas* sp. are effectively used as a biological control agent against plant pathogens [47]. Cellulase belongs to the family of hydrolytic enzymes reported for biocontrol of fungal pathogens [48]. In the present study, *B. subtilis* exhibited cellulase production which may partly explain fungal growth inhibition and reduced attack of wheat plants. Previous work reported that wheat rhizosphere bacteria showed the capacity to produce cellulase which contributes to the breakdown of fungal cell walls [49]. The indole acid produced by bacteria enhances plant growth and has a positive effect on yield of crop [50]. Our results showed that selected bacterial strains could improve wheat plant growth and protect them against *F. culmorum* attacks and they will minimize the use of chemical fertilizers and pesticides which are the main causes of disturbances in the climate such as global warming and climate change. In addition, it would be interesting to conduct multi-site trials in order to confirm the results obtained under controlled conditions and to have more detailed information on the PGPR and biocontrol effect of the selected strains.

Our examination of phenolic compound and chlorophyll contents showed limited variations when seedlings were infected with *F. culmorum* alone and an increase when wheat was inoculated with bacterial strains and this increase was correlated with protection by antagonists. In fact, phenolic metabolites (e.g., flavones, phenolic acids) are abundant plant compounds playing an important biological role in protecting plants against pests and pathogens [10,11,12,13,14,15,16,17,18,19,20,21,22,23,24,25,26,27,28,29,30,31,32,33,34,35,36,37,38,39,40,41,42,43,44,45,46,47,48,49,50,51]. Chlorophylls correspond to markers of stress response in plants, so fungal attacks could influence chlorophyll content. Some studies have shown that severe water stress on wheat allows the fall of growth and assimilation pigments (chlorophylls and carotenoids) but increases the content of phenolic compounds only in the shoots. [52]. Results presented here clearly indicate that *B. subtilis* (S6) and *B. amyloliquefaciens* (S8) have an antagonism against *F. culmorum*. In fact, these bacterial strains had the capacity to reduce fungal growth in vitro and protect wheat plants against Fusarium wilt. These results could be explained by the production of metabolites inhibiting mycelial growth and the induction of resistance in wheat against *F. culmorum*. These results corroborate several studies showing that metabolites produced by bacteria could trigger the plant’s defense response [53].

## 5. Conclusions

The intensive use of synthetic fertilizers and pesticides leads to harmful effects on the environment and human health, and therefore biological approaches such as the use of soil bacteria could be suggested to avoid further deterioration of ecosystems. Results presented in this study exhibited that the application *B. subtilis* and *B. amyloliquefaciens* separately improves wheat plants growth and protects them against attacks by *F. culmorum*. The use of these two bacteria in a biofertilization and biological control program seems encouraging and could reduce the rate of chemical fertilizers and pesticides. Therefore, it is an environment-friendly technology that can minimize soil pollution and maximize crop yields. Further studies are needed to determine not only the toxic antifungal metabolites produced by these strains, but also their potential use in future control strategies of wheat attacks by *F. culmorum* in fields.

## Figures and Tables

**Figure 1 microorganisms-11-01165-f001:**
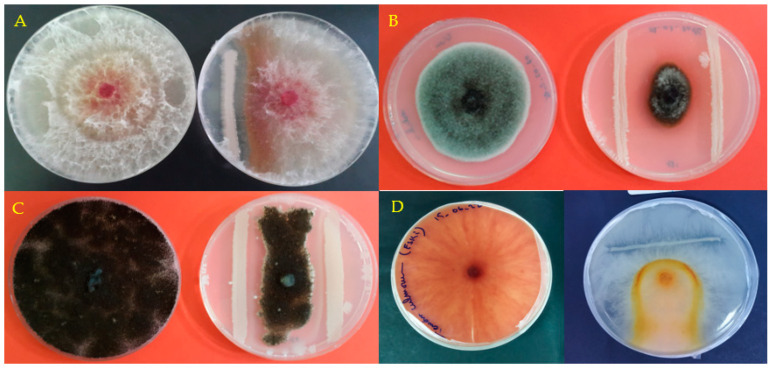
Effect of bacterial strain S8 on Fungal strains growth following 7 days of culture: *F. greminarium* (**A**), *Verticillium dahlia* (**B**), *Phytophtora* sp. (**C**) and *F. culmorum* (**D**). For each strain of fungus on the left the untreated control, on the right treated with the bacterial strain.

**Figure 2 microorganisms-11-01165-f002:**
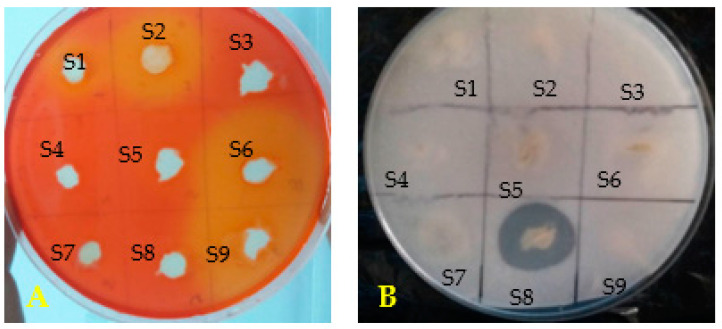
(**A**) Cellulase production, (**B**) Solubilization of inorganic phosphate.

**Figure 3 microorganisms-11-01165-f003:**
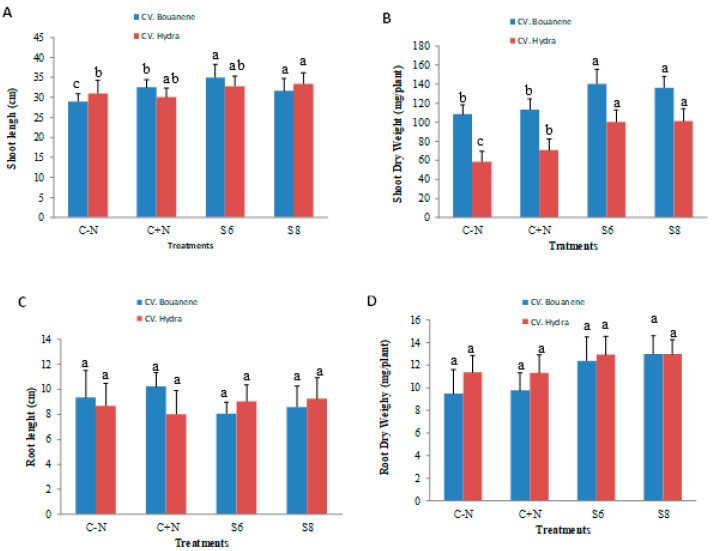
Effect of different bacterial inoculants (control, *Bacillus subtilis* (S6), *Bacillus amyloliquefaciens* (S8) on of wheat seedlings grown in pots containing soil. Plants irrigated with nutrient solution without nitrogen control (−) or irrigated with nitrogen control (+) Shoot length (**A**), shoot dry weight (**B**), root length (**C**) and root dry weight (**D**). Data are reported as mean values (*n* = 10). Columum with same color and different letters present no significant difference at 0.05.

**Figure 4 microorganisms-11-01165-f004:**
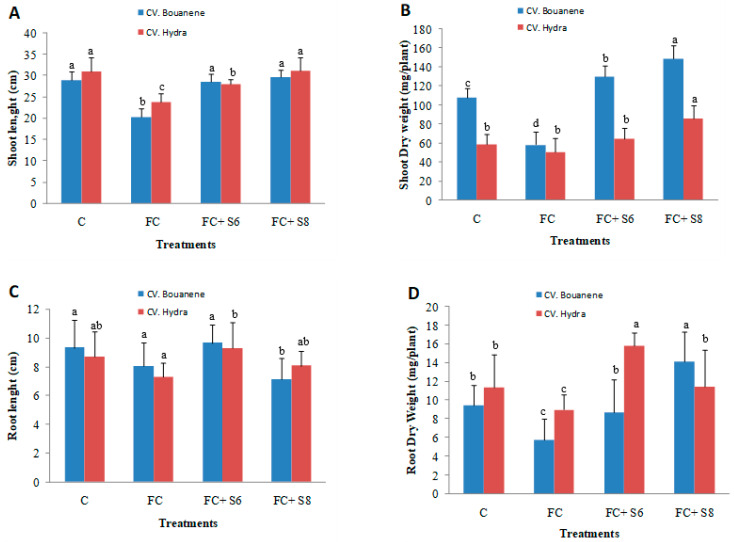
Effects of two bacterial strains (S6: *Bacillus subtilis* and S8: *Bacillus amyloliquefaciens*) on the growth of wheat seedlings treated by *F. culmorum* in pots. Shoot length (**A**), shoot dry weights (**B**), root length (**C**) and root dry weight (**D**). Data are reported as mean values (*n* = 10). Columum with same color and different letters indicate statistically significant difference (*p* < 0.05).

**Figure 5 microorganisms-11-01165-f005:**
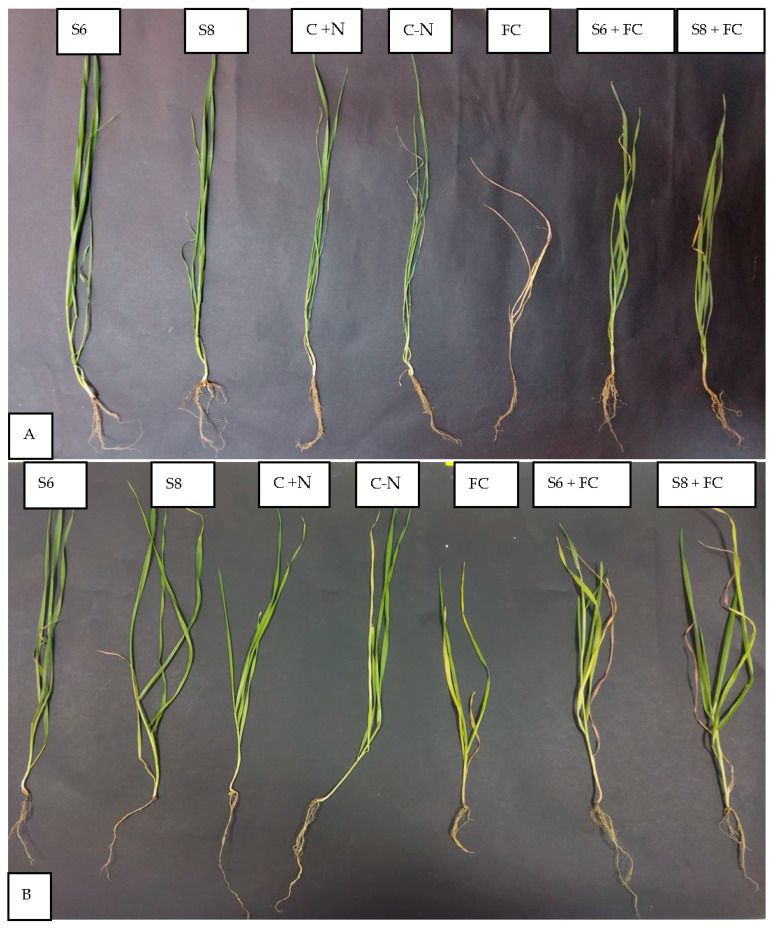
Seedlings of different treatments on day 30 post inoculation on shoots and root system of two wheat varieties plants (**A**): CV Bouanene; (**B**): CV Hydra. S6: plants inoculated only by bacterial strain S6; S8: plants inoculated only by bacterial strain S8; C + N: control treated with nitrogen; C-N: control without nitrogen; FC: plants inoculated with *F. culmorum;* FC + S6: plants inoculated with *F. culmorum* and the bacterial strain S6; FC + S8: plants inoculated with *F. culmorum* and the bacterial strain.

**Figure 6 microorganisms-11-01165-f006:**
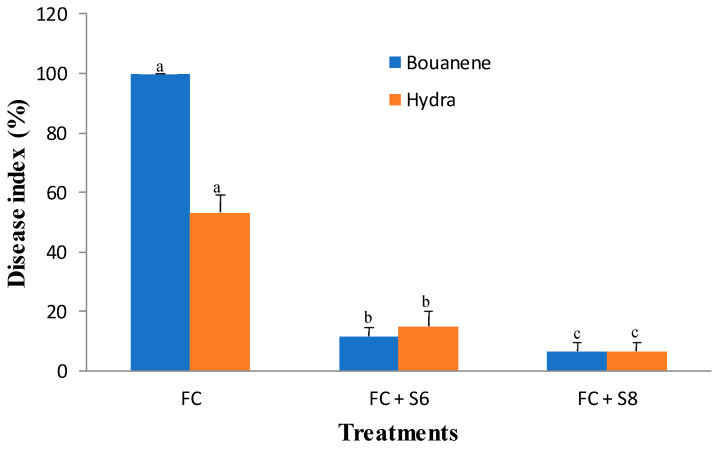
Disease index (±SD) of wheat (cultivars Bouanene and hydra) grown in pots with the phytopathogenic fungus *F. culmorum* and the antagonist bacteria. FC: wheat treated only with *F. culmorum*, FC + S6 wheat treated with *F. culmorum* and inoculated with bacterial strain S6, FC + S8: wheat treated with *F. culmorum* and inoculated with bacterial strain S8. Different letters indicate statistically significant difference (*p* < 0.05).

**Figure 7 microorganisms-11-01165-f007:**
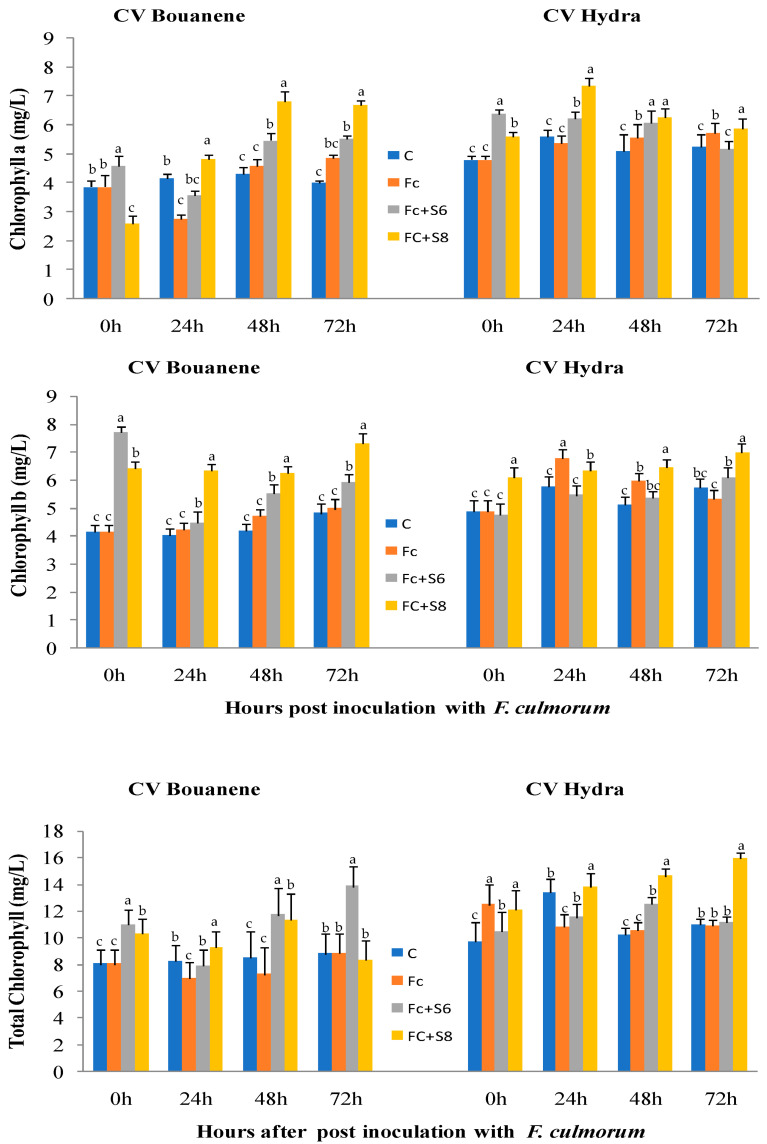
Effect of *F. culmorum* and antagonist bacteria on wheat leaf chlorophyll (B), contents (±SD). Different letters indicate statistically significant difference (*p* < 0.05).

**Figure 8 microorganisms-11-01165-f008:**
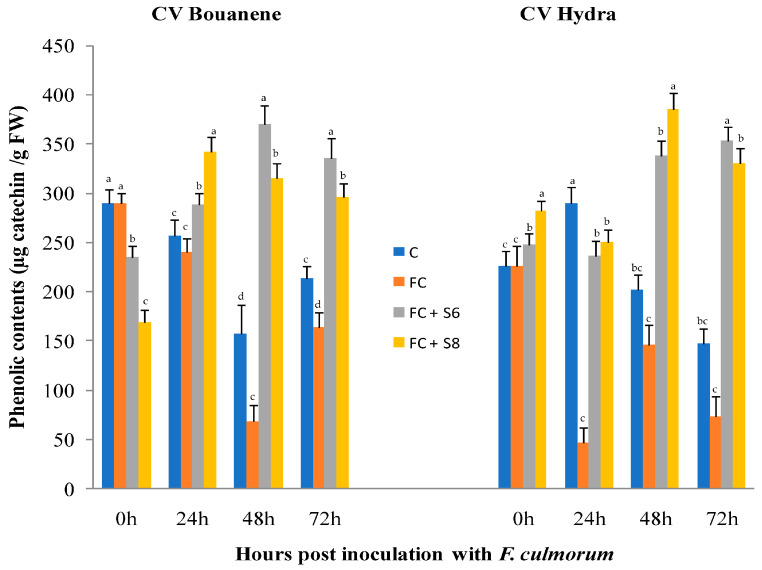
Effect of *F. culmorum* and antagonist bacteria on wheat leaf phenolic compounds contents (±SD). Different letters indicate statistically significant difference (*p* < 0.05).

**Table 1 microorganisms-11-01165-t001:** In vitro inhibition (%) of some bacterial isolates on mycelial growth of four phytopathogenic fungi. If there is one identical marker letter, the difference is insignificant, and if there is a different marker letter, the difference is significant. Lowercase letters indicate a significance level of 0.05.

	Inhibition of Fungal Growth (%)			
Strain No	*Fusarium graminearum*	*Verticilium dahlia*	*Phytophtora* sp.	*Fusarium culmorum*
S1	75.73 ± 0.67 ^a^	71.25 ± 2.17 ^b^	72.87 ± 2.93 ^a^	87.5 ± 1.78 ^a^
S2	0	45.00 ± 2.5 ^d^	0	80 ± 2.14 ^b^
S3	59.69 ± 2.42 ^bc^	0	48.84 ± 1.16 ^d^	57.5 ± 1.21 ^e^
S4	51.55 ± 1.78 ^c^	59.58 ± 0.72 ^c^	0	65 ± 0.98 ^d^
S5	65.50 ± 2.42 ^ab^	78.33 ± 0,72 ^a^	71.32 ± 5.37 ^a^	62.5 ± 1.25 ^de^
S6	64.19 ± 1.34 ^b^	77.08 ± 2.6 ^a^	64.34 ± 3.74 ^b^	72.5 ± 2.21 ^c^
S7	71.71 ± 1.78 ^a^	67.50 ± 4.33 ^b^	62.79 ± 2.53 ^b^	77.5 ± 1.98 ^bc^
S8	68.60 ± 3.08 ^a^	78.33 ± 0.72 ^a^	63.57 ± 3.36 ^b^	84.5 ± 1.54 ^ab^
S9	61.24 ± 6.71 ^b^	75.83 ± 3.82 ^a^	68.99 ± 1.34 ^ab^	14.5 ± 0.25 ^f^

**Table 2 microorganisms-11-01165-t002:** Specific properties of the selected bacteria.

Strains Reference	NH_3_ (mg/mL)	IAA (mg/mL)	Solubilization of Inorganic Phosphate (mg/mL)	Mineralization of Organic Phosphate (mg/mL)	Nitrogen Fixation	Siderophore Production	Cellulase
S1	0.16 ± 0.01	0.002 ± 0.002	0	0	-	-	-
S2	0.11 ± 0.01	0	0	0	+	-	-
S3	0.15 ± 0.01	0.007 ± 0.003	0.14 ± 0.04	0	+	-	-
S4	0.3 ± 0.07	0	0	0	+	-	-
S5	0.18 ± 0.03	0.007 ± 0.004	0.04 ± 0.002	0	+	-	-
S6	0.69 ± 0.05	0.21 ± 0.01	0.14 ± 0.02	0	+	+	+
S7	0.25 ± 0.04	0	0.06 ± 0.01	0	-	-	+
S8	0.39 ± 0.02	0.002 ± 0.001	0.15 ± 0.05	0.003 ± 0.02	+	+	-
S9	0.02 ± 0.01	0	0	0	-	-	-

**Table 3 microorganisms-11-01165-t003:** Characterization of two bacterial isolates (S6 and S8) using 16S rDNA.

Bacterial Isolates	Classification
S6	*B. subtilis*: 168 (NR_102783.1)
S8	*B. amyloliquefaciens:* 9 MPA 1034 (NR_117946.1)

## Data Availability

Not applicable.

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
