# Peer review of "Biocontrol Potential of Some Rhizospheric Soil Bacterial Strains against Fusarium culmorum and Subsequent Effect on Growth of Two Tunisian Wheat Cultivars"

_microorganisms, 2023, doi:10.3390/microorganisms11051165_

Round 1
Reviewer 1 Report
The discovery of new PGP bacterial strains is very important for the development ecological technology of agronomy. The study in the article was carried out in a complex manner. Showing the main properties for PGP bacteria.
I have a few comments:
Table 1. it is not clear how the percentage of inhibition of fungal growth was determined
Fig. 2. (B) Solubilization of inorganic phosphate: shown only for S8, but in table 2 solubilization for strain S3, S5-S8.
In accordance with Table 2, only strain S6 can be said with certainty that it produces IAA, the rest are found to be too small and are within the margin of error.
Fig 5. Verify signatures - part lost
Fig 6. Not explained the image of "emptiness" in column C: C: Wheat grown without fungus - did not grow at all?
Fig 7. Verify signatures - part lost
There are abbreviations PGP, PGPR and PGPB in the article, it is necessary to give a transcript and bring to uniformity
Author Response
Table 1. it is not clear how the percentage of inhibition of fungal growth was determined
Title modified and inhibition calculation was explained in Material and methods section
Fig. 2. (B) Solubilization of inorganic phosphate: shown only for S8, but in table 2 solubilization for strain S3, S5-S8.
The spectrophotometric technique is more sensitive and makes it possible to quantify the solubilization of phosphate
In accordance with Table 2, only strain S6 can be said with certainty that it produces IAA, the rest are found to be too small and are within the margin of error.
Sentence was modified in the text according to results presented in table 2 and suggested by reviewer
Fig 5. Verify signatures - part lost
Figure 5 verified and updated
Fig 6. Not explained the image of "emptiness" in column C: C: Wheat grown without fungus - did not grow at all?
No symptoms observed on wheat plants and this column is removed from figure 6
Fig 7. Verify signatures - part lost
Figure 7 verified and updated
Reviewer 2 Report
This article presented Biocontrol potential of some rhizospheric soil bacterial strains against Fusarium culmorum and subsequent effect on growth of two Tunisian wheat cultivars. The study is well organized and data is well arranged. The findings would be helpful for future studies. Before recommending this article for publication, there are some shortcomings for that should be resolve.
Overall, the study is well designed and presented in a good way, but mostly the literature is not cited. Grammatical and typos must be revised.
Check and Correct the bacteria names in the whole MS. Most of the time . is missing between generic name and sp name.
The sentences are very long and vague.
Discuss main findings of the study quantitatively.
Also discuss brief methods.
The section is well described however further information is required such as significance of rhizobacteria as bio fertilizers, its mechanism, harm and losses by phytopathogens specifically to wheat.
Line 47 lack reference and should be cited with recent study. https://doi.org/10.3390/molecules27196281
Line 53-55 the sentence is not clear and wrong grammatically.
At the end of the introduction state hypothesis of the study and aim of the study.
Also discuss novelty of the study.
Line 64 word Indeed should be replacing with appropriate word.
Section 2.3 should be cited with recent study https://doi.org/10.3390/microorganisms10050954
Most of the text species names are not italicized. Please italicize all species or bacteria names.
Figure 7 and 8 add statistical analysis to find significant difference among the treatments.
All obtained results should be discussed and compared with relevant studies.
In conclusion add study gap and future perspective.
Author Response
Overall, the study is well designed and presented in a good way, but mostly the literature is not cited. Grammatical and typos must be revised.
The references are updated in the text
Check and Correct the bacteria names in the whole MS. Most of the time. is missing between generic name and sp name.
Names of bacteria are checked
The sentences are very long and vague
modified
Discuss main findings of the study quantitatively.
Discussion is amended
Also discuss brief methods.
The section is well described however further information is required such as significance of rhizobacteria as bio fertilizers, its mechanism, harm and losses by phytopathogens specifically to wheat.
As suggested by reviewer the manuscript is updated
Line 47 lack reference and should be cited with recent study. https://doi.org/10.3390/molecules27196281
Reference is added
- Sadiqi, S.; Hamza, M.; Ali, F.; Alam, S.; Shakeela, Q.; Ahmed, S.; Ayaz, A.; Ali, S.; Saqib, S.; Ullah, F.; Zaman, W. Molecular Characterization of Bacterial Isolates from Soil Samples and Evaluation of their Antibacterial Potential against MDRS. Molecules 2022, 27, 6281. https://doi.org/10.3390/molecules27196281
Line 53-55 the sentence is not clear and wrong grammatically.
At the end of the introduction state hypothesis of the study and aim of the study.
Also discuss novelty of the study.
done
Line 64 word Indeed should be replacing with appropriate word.
done
Section 2.3 should be cited with recent study https://doi.org/10.3390/microorganisms10050954
Reference cited
Most of the text species names are not italicized. Please italicize all species or bacteria names.
Species and bacteria names are checked and modified
Figure 7 and 8 add statistical analysis to find significant difference among the treatments.
Letters are added for significant difference
All obtained results should be discussed and compared with relevant studies.
Discussion updated using new references of relevant studies
In conclusion add study gap and future perspective.
Conclusion amended
22. Abid, S.; Farid, A.; Abid, R.; Rehman, M.U.; Alsanie, W.F.; Alhomrani, M.; Alamri, A.S.; Asdaq, S.M.B.; Hefft, D.I.; Saqib, S.; Muzammal M , Abdelrahman Morshedy S,. Alruways MW and Ghazanfar S (2022) Identification, Biochemical Characterization, and Safety Attributes of Locally Isolated Lactobacillus fermentum from Bubalus bubalis (buffalo) Milk as a Probiotic. Microorganisms 2022, 10, 954. https://doi.org/10.3390/microorganisms10050954
Reviewer 3 Report
The authors investigated the biocontrol potentials of soil bacterial strains on Fusarium culmorum as well as the effects on wheat cultivars. I have some comments to suggest author modify their manuscript.
First, the English writing in the whole manuscript should be checked and improved, for example, in line 21, in pot experiments, in controlled conditions. line 21 to 23, a long sentence is not appropriate. Line 259 F. culmorum should be italic,
In the introduction section, line 45 to 52, the background of bio control of pathogens should be extended, related references can be checked (DOI: 10.1016/j.tifs.2022.04.002).
Figure 1, the quality of presented figure should be improved, for example B and D were not well aligned.
For the references, most of them were before 2010, it should be checked and updated in the last 10 years.
In all, novel bacterial strains were identified for managing F. culmorum, and this manuscript could be accepted but need revision.
Author Response
First, the English writing in the whole manuscript should be checked and improved, for example, in line 21, in pot experiments, in controlled conditions. line 21 to 23, a long sentence is not appropriate. Line 259 F. culmorum should be italic,
Abstract updated and species names checked
In the introduction section, line 45 to 52, the background of bio control of pathogens should be extended, related references can be checked (DOI: 10.1016/j.tifs.2022.04.002).
Reference cited and introduction section updated
Figure 1, the quality of presented figure should be improved, for example B and D were not well aligned.
Figure checked
For the references, most of them were before 2010, it should be checked and updated in the last 10 years.
References used in introduction and discussion sections are checked and updated as requested